# Hemicellulose Films from Curaua Fibers (*Ananas erectifolius*): Extraction and Thermal and Mechanical Characterization

**DOI:** 10.3390/polym14152999

**Published:** 2022-07-25

**Authors:** Mariana Roldi-Oliveira, Layse M. Diniz, Anastasia L. Elias, Sandra M. Luz

**Affiliations:** 1Mechanical Department, Technology Faculty, University of Brasília, Brasília 70910-900, Brazil; mroldi.arq@gmail.com (M.R.-O.); laysemendis@hotmail.com (L.M.D.); 2Department of Chemical and Materials Engineering, University of Alberta, Donadeo Innovation Centre for Engineering, Edmonton, AB T6G 1H9, Canada; aelias@ualberta.ca

**Keywords:** hemicellulose, curaua fibers, experimental design, thermal characterization

## Abstract

With growing environmental concerns over synthetic polymers, natural polymeric materials, such as hemicellulose, are considered a good sustainable alternative. Curaua fibers could be an excellent source of biopolymer as they have a relatively high hemicellulose content (15 wt%) and only a small amount of lignin (7 wt%). In this work, hemicellulose was extracted by an alkaline medium using KOH and the influence of the alkali concentration, temperature, and time was studied. A hemicellulose film was produced by water casting and its mechanical, thermal, and morphological properties were characterized. The results show that the best method, which resulted in the highest hemicellulose yield and lowest contamination from lignin, was using 10% (*w*/*v*) KOH concentration, 25 °C, and time of 3 h. The hemicellulose film exhibited better thermal stability and elongation at break than other polymeric films. It also exhibited lower rigidity and higher flexibility than other biodegradable polymers, including polylactic acid (PLA) and polyhydroxybutyrate (PHB).

## 1. Introduction

The constant growth in demand for polymers is a concern since the production of synthetic polymers involves petroleum, and the degradation of synthetic polymers in the environment has numerous negative effects [1]. To mitigate this problem, research related to natural polymers has achieved prominence in the scientific community. In addition to being obtained from renewable sources, many natural polymers more easily degrade in nature than synthetic ones. Some examples of natural polymers (which are also called biopolymers) are hemicellulose, polylactic acid (PLA), and polyhydroxybutyrate (PHB).

Currently, PLA and PHB are among the most common natural polymers used in the industry, for example, in the manufacture of biodegradable food packaging [2]. These materials exhibit high oxygen barrier performance, good water resistance, and are transparent [3]. However, properties such as low hardness, high stiffness, high brittleness, and low crystallization rate limit their commercial applications [4].

Hemicellulose polymers are edible, biodegradable, and environmentally friendly and, because of their specific physicochemical properties, which include high molecular weight, water solubility, and absence of toxicity, they can be used as food packaging and biomedical materials [5]. Hemicelluloses are generally comprised of 80–200 units of sugar residues and can be grouped according to their sugar type, namely hexose (glucose, mannose, and galactose), pentoses (xylose, arabinopyranose, and arabinofuranose), hexuronide acid (glucuronic acid, methylglucoronate, and galacturonate), and deoxyhexose (rhamnose and fructose) [6]. Hemicellulose polymers can have their chains formed by a single monosaccharide or more units depending on the type of plant tissue and the plant species to which it belongs [7]. This polymer is abundant in nature and can be extracted from natural fibers [8].

In this context, curaua (*Ananas comosus var. erectifolius*), a monocotyledonous plant native to the Amazon region, similar to the pineapple plant, could be a good source of hemicellulose [9]. As for all lignocellulosic materials, curaua comprises cellulose, hemicellulose, and lignin. The fibers used in this research have 61.87% cellulose, 15.02% hemicellulose, 6.83% lignin, 5.80% of moisture, 0.56% of ash, and 9.92% of other undetermined components [9]. Curaua has a small lignin content and this factor can facilitate the access of chemical reagents to hemicellulose. Sugarcane bagasse fibers, for example, have 43.36% cellulose, 23.27% hemicellulose, and 21.38% lignin [10]. Despite having a higher percentage of hemicellulose compared to curaua fibers, the comparatively high percentage of lignin makes the hemicellulose extraction process difficult.

Hemicellulose is branched and hydrophilic [11], strongly bonded to cellulose microfibrils, and provides stability and flexibility to vegetable fibers [12]. Like lignin, hemicellulose is an amorphous and essentially thermoplastic polymer, for which the main softening point is the glass transition temperature [13].

Hemicellulose can be used as a biofilm for food packaging [14]. The low oxygen permeability, good mechanical strength, and high flexibility of this material are important properties for such applications. Hemicellulose can therefore be a sustainable alternative to synthetic plastics, which are now widely used as food packaging materials [14,15].

However, despite the abundance of hemicellulose, it is still not heavily studied. The main bottleneck to the widespread application of hemicellulose is the challenge of separating this material from the other macromolecular components of the lignocellulosic fibers while preserving its polymeric characteristics [16]. As the polymer is bound to cellulose and lignin in the plant cell wall, isolating procedures are required to separate these components from the vegetable raw material [14], while preserving the hemicellulose structure. High concentrations of alkaline solution, which are generally used to separate cellulose, can depolymerize hemicellulose [17].

The two main alkaline solvents used to extract hemicellulose are aqueous solutions of sodium hydroxide (NaOH) and potassium hydroxide (KOH) [11,18]. The KOH solution was found to be more effective than NaOH for the recovery of hemicellulose from a variety of plants. Furthermore, one of the main methods of obtaining the polymer is by fractional precipitation with acidification and the addition of organic solvents such as ethanol, methanol, or acetone. Thus, the alkaline solutions are generally neutralized with acetic acid and treated with an excess solvent [7].

This research aims to develop an optimized method to extract hemicellulose from curaua fibers. The influence of the three main factors for the hemicellulose extraction procedure—namely KOH concentration, temperature, and time—on both the yield and characteristics of the obtained material are also investigated herein. Furthermore, after the film formation, the main properties of the material, including the mechanical properties and morphology, are characterized.

The development of an efficient methodology for extracting hemicellulose is an important step towards developing biodegradable polymer films. Here, the physicochemical characteristics of the materials are explored to evaluate the suitability of these films for a wide number of applications.

## 2. Materials and Methods

### 2.1. Materials

Untreated curaua fibers were provided by the Support Center to Community Action Projects (CEAPAC), Santarém, PA, Brazil in long bundles of 80 cm. These bundles were post harvested and dried. The chemical reagents used in this work, including potassium hydroxide (85% purity), glacial acetic acid, and ethanol (95% *v*/*v*), were received from Sigma, São Paulo, SP, Brazil. For a comparison of the dynamic mechanical properties, the biopolymer PHB was obtained from BRS Bulk Bio-Pellets, Bulk Reef Supply Golden Valley, MN, USA and a PLA was received from NatureWorks LCC, Minnetonka, MN, USA.

### 2.2. Hemicellulose Extraction

The extraction methodology was adapted from Bahcegul et al. [19]. A 2^3^ factorial design was implemented, where the three independent variables were the KOH solution concentration, temperature, and reaction time. A high and low value of each independent variable was selected. The effect of these variables on the final weight of the hemicellulose was determined; this value was also used to calculate the overall yield with respect to the initial dry weight of in natura curaua. Experiments were performed in duplicate and the levels determined for each factor are given in Table 1.

The tests were randomized to avoid biased results and increase the efficiency of the analyses. The significance of the effects was determined with 95% confidence using the Student’s t distribution with eight degrees of freedom [20].

Each test used 10 g of in natura curaua fibers immersed in 200 mL of distilled water at 25 °C for 1 h. After filtration, the treated fibers were added to 100 mL of KOH (10 or 20% *w*/*v*) solution and stirred for 3 or 5 h at 25 or 50 °C, according to the conditions described in Table 1. After another filtration, the pH of the liquor was adjusted to 4.8 using acetic acid. To separate the insoluble fraction, the liquor was centrifuged at 4000 rpm for 5 min at 25 °C. Then, 250 mL of a 1:10 solution of glacial acetic acid: ethanol was added to the liquor and the solution was kept at 25 °C for 24 h, allowing the hemicellulose to precipitate. After that, the precipitated hemicellulose was filtered and dried at 25 °C for 48 h and then dried in the oven at 60 °C until a constant weight was reached.

### 2.3. Hemicellulose Water Casting

After the optimal parameters were identified using the procedure described in Section 2.2, 230 g of curaua fibers were treated to obtain hemicellulose. A film was produced via the water casting of the hemicellulose solubilization in distilled water at a concentration of 33% (*w*/*v*). The solution was stirred for 2 h at 35 °C, centrifuged at 4000 rpm for 10 min and deposited on the glass base (310 × 210 mm in dimension) for molding, as shown in Figure 1. The hemicellulose was dried at room temperature for eight days, resulting in the formation of a film with 0.3 mm thickness.

### 2.4. Thermogravimetry (TG) and Differential Scanning Calorimetry (DSC)

Curaua fiber, hemicellulose, and the residual fibers obtained from the factorial design experiments were analyzed in a TG/DSC simultaneous thermal analyzer, model SDT Q600 (TA Instruments, New Castle, DE, USA) in alumina sample pans under nitrogen atmosphere (N_2_) flowing at a rate of 50 mL/min. A heating rate of 10 °C/min was used to heat the samples from 25 °C to 950 °C.

### 2.5. Coupled Thermogravimetry–Fourier Transform Infrared Spectroscopy (TG-FTIR) Analysis

The gases released under TG during the thermal degradation of the 16 hemicellulose samples (from eight different methods) prepared following the factorial design were collected by a Nicolet iS10 spectrometer coupled to the thermal analyzer (Thermo Fisher Scientific, Carlsbad, CA, USA). The spectra were collected in the range from 4000 to 400 cm^−1^ with a resolution of 4 cm^−1^ and 128 scans. The temperatures in the transfer line and the gas cell were 195 and 200 °C, respectively.

### 2.6. Scanning Electron Microscopy (SEM)

The hemicellulose films produced using the factorial design were sputtered with a thin layer of gold and analyzed in a JEOL scanning electron microscope, model JSM-7001F, Peabody, MA, USA with an acceleration voltage of 15 kV using the secondary electrons system. To determine the chemical composition of the samples, some regions were analyzed by SEM with energy-dispersive X-ray spectroscopy (EDS).

### 2.7. Molecular Weight of Hemicellulose Films by Gel Permeation Chromatography (GPC)

The molecular weight and dispersity (Đ) of the hemicellulose prepared using the conditions that provided the highest yield were characterized using gel permeation chromatography (GPC) (1260 Infinity Multi-Detector GPC/SEC System, Agilent Technologies, Santa Clara, CA, USA). The GPC system included viscosity refractive index and light-scattering detectors. Two columns (2 Agilent PL aquagel-OH MIXED-H) connected in series were used to obtain a better resolution and increase the instrument′s detection range.

The system was calibrated using poly (ethylene oxide) standards provided by Agilent Technologies. Water containing 0.2 M sodium nitrate (NaNO_3_) was used as a mobile phase for the analysis and 1 mL/min flow rate at 30 °C for the column oven temperature.

### 2.8. Quartz Crystal Microbalance (QCM)

A Quartz crystal microbalance (QCM) is an extremely sensitive mass balance that measures the nanogram-to-microgram level changes in mass per unit area. The model used for the analysis was Stanford Research Systems′ QCMeletronics (QCM200, Gainesville, GA, USA). Gold-coated quartz crystals and a frequency of 5 MHz were used. The crystal holder was immersed in a 400 mL polypropylene beaker containing 300 mL of DI water and was held in a fixed vertical position perpendicular to the solution′s surface during all measurements [21]. The experiments were performed in deionized water and carried out at room temperature (22 ± 0.3 °C). Through the software provided by the supplier, the frequency of the crystal was recorded as a function of time.

### 2.9. Tensile Tests

For tensile tests, the specimens were cut into rectangular strips that were 300 mm in total length, 17 mm in width, and 0.30 mm in thickness, according to the ASTM D882-02 standard [22]. The tests were performed at 24 °C and 62% relative humidity using a Universal Instron EMIC 23-5D test machine (Norwood, MA, USA), 50 N load cell, and 5 mm/min speed.

### 2.10. Dynamic Mechanical Analysis (DMA)

The dynamic mechanical properties of the hemicellulose film were tested using a dynamic mechanical analyzer (DMA 8000, PerkinElmer, Inc, Waltham, MA, USA). For comparison, the PLA and PHB films were prepared by compression molding at a temperature of 175 °C and were analyzed by DMA [23]. Flat specimens (20 mm in total length, 9.5 mm in gauge length, 5 mm in width, and 0.26 mm in thickness) were analyzed in tension mode as they were heated from 20 to 180 °C at 3 °C/min. Samples were displaced to 0.05 mm at a 1 Hz frequency.

## 3. Results and Discussion

### 3.1. Best Conditions from 2^3^ Factorial Design

For each set of conditions, the total average hemicellulose yield (in grams) and the calculated yield (the final weight of the hemicellulose divided by the initial weight of dried curaua fibers, in percent) is given for each experiment. The effects, the second Student’s t distribution with eight degrees of freedom to 95% confidence, a standard error of 0.11, and their values are all shown in Table 2.

Pieta and Bariccatti [24] studied the extraction of sugars from sugarcane bagasse from chemical pretreatment with diluted sulfuric acid, and a factorial design that included three factors with two levels each. The interaction among the factors (concentration/temperature, and mass) presented lower values than the minimum necessary to be significant. Only the concentration was found to have a significant effect on the yield of sugars (as determined by the refractive index measurement and high performance liquid chromatography), with 10% acid concentration being more effective than 5%. Thus, the 2^3^ factorial design indicated that the concentration is the only factor with a significant influence over the hemicellulose yield for the parameters studied. Here, as shown Table 2, six of the eight methods (75% of the samples) were found to have higher extracted product yield than the original amount of hemicellulose in curaua fibers (15.02%, [9]). For example, in sample 8 (20%_50 °C_5 h), the yield is almost 39% higher than the percentage of hemicellulose expected in curaua. This high value may result from the final sample containing components such as cellulose, lignin, and hemicellulose, as was also observed by Rodrígues et al. [9]. This can be justified by the fact that the main components of the curaua fiber (lignin, hemicellulose, and cellulose) present different reactivities in response to alkaline treatments. As a base, KOH acts to solubilize hemicellulose, enabling its extraction [25]. However, the higher than expected product yield suggests that additional components were also removed. Because hemicellulose and lignin form complexes, lignin may be extracted, bound to hemicellulose. While cellulose generally shows good resistance to alkaline treatment due to its crystalline structure [26], it is also possible that some cellulose is extracted, particularly at high KOH concentrations. These possibilities will be explored in subsequent sections [9,26,27].

### 3.2. Thermal Characterization of Hemicellulose from 2^3^ Factorial Design Characterization: TG-DTG/DSC

The hemicellulose extracted by the different methods was characterized by thermogravimetry and its derivative (TG-DTG) and differential scanning calorimetry (DSC). These methods can provide information about the thermal stability of the materials, glass transition temperature (T_g_), as well as the possible impurities on the hemicellulose surface. The results of this analysis can be seen in Figure 2. From Figure 2A,B, one can observe that all samples presented the same behavior, indicating four stages of weight loss. The first is very subtle and refers to moisture loss (up to 150 °C); the second stage (from 190 to 325 °C) is represented by two peaks related to hemicellulose degradation [28]; the third stage (from 350 to 450 °C) results from cellulose degradation; and the fourth stage (from 725 to 875 °C) is related to the residues of the KOH solution, with its possible conversion to oxide [29].

The degradation peaks of hemicellulose occurred at the temperatures shown in Table 3. The first peak of hemicellulose degradation occurred at 227 °C (average), with a maximum weight loss rate of 0.60%/°C (method 1: 10% _25 °C _3 h); the second peak occurred at 273 °C (average) and showed a higher degradation intensity than the first stage with degradation rates between 0.60 and 1.04%/°C. The weight loss in the second stage, which mostly results from hemicellulose degradation, corresponded to 41–49% weight loss. Typically, hemicellulose degradation occurs at a temperature range between 220 and 315 °C [30], which is very close to the range observed in this work (190 and 325 °C).

Cellulose degradation typically occurs in the temperature range of 315–400 °C [30]. In the 350–500 °C and 725–875 °C ranges, the two subsequent peaks correspond to cellulose and residues of the K_2_O (KOH converted into oxide), respectively. However, the degradation temperature of the cellulose content in the samples was almost 100 °C higher than expected. This occurrence may result from the predominant presence of hemicellulose in the samples, delaying the transfer of heat to the cellulosic innermost portions of the samples, leading to the late degradation of this polymer.

For all hemicellulose samples extracted with a higher concentration of KOH (20% (*w*/*v*)), the initial cellulose degradation temperatures were lower than for the samples carried out under the same temperature and time conditions and the lower KOH concentration (10% *w*/*v*). This may indicate that KOH 20% (*w*/*v*) leaves more KOH residues (ashes) in the samples. The presence of these residues may be causing the breakdown of chemical bonds in the cellulose, thus reducing the initial temperature of cellulose degradation [30]. For the KOH residues, the lowest peaks correspond to the samples extracted with 10% (*w*/*v*) while the samples extracted with a concentration of 20% (*w*/*v*) showed higher peaks.

The thermal stability of all samples was similar, presenting an average of 228 °C. At 900 °C, the hemicellulose extracted with a concentration of 10% (*w*/*v*) presented a higher residue content than the samples extracted with a concentration of 20% (*w*/*v*).

To obtain more information on the effects of the extraction of the hemicellulose from the fibers, the in natura curaua fibers and the extracted fibers were also studied. Figure 2C,D shows the thermal characterization of the fibers from which the hemicellulose was extracted using the most extreme extraction conditions: method 1 (10%_25 °C_3 h) and method 8 (20%_50 °C_5 h).

In the DTG curve for the in natura fiber, fluctuation occurred from 250 °C followed by a large peak at 350 °C, and another undulation at roughly 500 °C. Such occurrences are related to the degradation of hemicellulose, cellulose, and lignin [30].

In each set of extracted fibers, the cellulose remaining after hemicellulose removal was degraded at temperatures much lower than its natural degradation range (315 and 400 °C), which indicates that hemicellulose in the curaua fibers increases its thermal stability, and/or that the structure of the cellulose in the fibers is damaged by the alkaline treatment. Furthermore, for the fibers treated by methods 1 (10%_25 °C_3 h) and 8 (20%_50 °C_5 h), a weight loss of 52% and 38% occurred between 200 and 330 °C (as shown in Figure 2C,D). This shows that in the fibers treated with the higher concentration of KOH (20% (*w*/*v*)), there was a lower amount of residual cellulose than those treated with 10% (*w*/*v*), which indicates that in the methods with 20% (*w*/*v*) concentration, cellulose portions were extracted from the fibers along with hemicellulose.

In the temperature range of 727–850 °C, there was a small weight loss (6.1%) in the fiber treated by method 1 (10%_25 °C_3 h). Between 700 and 920 °C, there was a large weight loss (22.2%) in the fiber treated by method 8 (20%_50 °C_5 h), while the in natura fiber did not present any degradation in this temperature range. This means that the degradation that occurs in the treated fibers is related to the amount of KOH used in each extraction method, as for the samples for which the higher the concentration of KOH was used, there is a sharp and large weight loss, which was attributed to the breaking of intermolecular hydrogen bonds [31].

DSC curves for hemicellulose samples prepared using each method are shown in Figure 3; four thermal events are observed for each sample. The first event was endothermic with low energy occurring at 190 °C, on average, shortly before the hemicellulose degradation range. This peak is attributed to hemicellulose glass transition (T_g_), which occurs from 150–220 °C [14,32]. As the temperature increases to between 200 and 250 °C, an undulation occurs. This is the cellulose glass transition temperature [33].

The third event is represented by a wider and more intense exothermic peak in the temperature range between 250 and 330 °C, which signals the hemicellulose degradation [34]. This thermal event appears as a wide endothermic wave without characteristic peaks. This event is attributed to the degradation of the cellulosic portion present in the samples [34]. It is observed that, for this event, the sample with 20%_50 °C_5 h was the only one that presented a definite peak at 375 °C, indicating the need for higher energy in the degradation process and thus confirming the presence of a higher amount of cellulose (which was subjected to the most aggressive conditions of any of the samples).

Hemicellulose is comprised of various saccharides (xylose, mannose, glucose, galactose, etc.). It typically has random, amorphous structured branches, which are very easy to remove from the main stem and that easily degrade to volatiles (CO, CO_2_, and some hydrocarbon) at low temperatures [30]. Between 700 and 800 °C, another exothermic event could be relative to the residual K_2_O of the extraction solution in the hemicellulose samples. At higher temperatures, some low energy undulation is observed in the curves of each sample, which may be due to the chemical reactions of carbonaceous, extractive residues, etc., in the samples.

### 3.3. Coupled Thermogravimetry–Fourier Transform Infrared Spectroscopy (TG-FTIR)

Simultaneous TG and FTIR analyses were performed to enable the real-time FTIR characterization of the gases released during weight loss induced by the heating of the hemicellulose films. Figure 4 presents the spectrum of the hemicellulose sample extracted at 10% _25 °C_3 h, indicating the main components released during the degradation of the samples. As expected, water and CO_2_ are the two main degradation products. During the analysis, the spectra with higher absorbance intensity are obtained around the DTG peaks and the hemicellulose degradation range. All samples showed similar profiles, but the absorbance intensity of the peaks varied.

The peaks relative to the release of CO_2_ (2350 and 650 cm^−1^) and H_2_O (3900 and 3500 cm^−1^) may be due to the degradation of the three main components of the fibers: cellulose, hemicellulose, and lignin [35].

Small peaks occurring at 1520 cm^−1^ (phenylpropane skeleton); 1460 cm^−1^ (C–H deformation combined with aromatic ring vibration); and 1330 cm^−1^ (syringyl ring breathing with C–O elongation) are usually attributed to lignin [36]. However, such peaks were not observed here, confirming the result previously indicated by the thermal analysis that there is no lignin present in the hemicellulose samples.

As can be seen in Table 4, the extractions carried out with a concentration of 10% (*w*/*v*) showed higher absorbance intensity than the samples under the same conditions of time and temperature, but with a concentration of 20% (*w*/*v*) in bands 2915 cm^−1^, 1730 cm^−1^, and 1250 cm^−1^ which correspond, respectively, to the release of CH_4_ from the hemicellulose ether group, C = O from the hemicellulose ether group, and the C–O–C bond [37]. This indicates that the concentration of hemicellulose was higher in the samples extracted with the KOH solution at its lower level (10% *w*/*v*). In band 1243 cm^−1^, another peak related to a C–O–C bond occurred, but this was from the cellulose chain [38]. This peak′s low intensity confirms the relatively low cellulose concentration in the samples.

### 3.4. Hemicellulose Film Characterization

The previous results show that method 1 (10%_25 °C_3 h) more effectively extracted the hemicellulose from curaua fibers, as described in Section 2.3 (Methodology). Films prepared using this method were characterized using QCM, GPC, scanning electron microscopy (SEM), tensile and dynamic mechanical analysis (DMA), and the results are discussed herein.

QCM was used to monitor the behavior of the hemicellulose upon immersion in water. The oscillation frequency can be affected by adding or removing small amounts of hemicellulose mass on the electrode surface [21]. By QCM, rapid drops in mass were observed within 10 min of immersion in water, indicating that the samples were quickly dissolving and/or delaminating from the substrate. This is consistent with our observations by eye, where we noted that the samples immersed in water tend to dissolve within 6 min. It can be inferred that this sudden drop is related to the dissolution of hemicellulose as it is completely soluble in water [39].

From the molar mass distribution curve determined by GPC, the weight-average (Mw) and number-average (Mn) molecular weight were obtained, as well as the polydispersity (Mw/Mn). The Mw was found to be 118,000 g/mol, the Mn was determined to be 73,371 g/mol, and the polydispersity was 1.61.

Hemicellulose is a heterogeneous glycan with polydispersity. The greater the polydispersity coefficient (Mw/Mn), the greater the hemicellulose molecular weight distribution [40]. The polydispersity found for hemicellulose with 1.61 indicates that the polymer has a relatively uniform particle distribution, as was also observed by Xu et al. [41].

The average weighted molecular weight and number-average molecular weight values indicate that the hemicellulose obtained has a low molecular weight. Hemicelluloses are branched polymers of low molecular weight with a typical degree of polymerization (DP) of 80–200 [42]. In their studies, Ding et al. [43] observed the number average molecular weight (Mw) values in the range of 42,900–44,200 Da for hemicelluloses obtained with different extraction methods. These values may be lower than those found here since the source plant was different (switchgrass).

The scanning electron microscopy of the surface of hemicellulose films is shown in Figure 5. We noticed the hemicellulose film has a typical morphology of polymeric materials. The film has a dense structure, with some irregularities and with low porosity; a similar structure was also seen by da Silva Braga and Poletto [44]. The dense structure of films with reduced surface porosity is due, in large part, to the particle size and the resulting rigid hydrogen bonded network in the biopolymer matrix [45]. This corroborates the result found for GPC analysis, in which the polydispersity for cellulose is characteristic of a material with a good molar distribution.

Some particles on the film surface are observed and these were identified using energy dispersion X-ray spectroscopy (EDS) as 3.91% carbon, 58.67% oxygen, 24.95% potassium, and 12.48% gold. In addition to the gold present in the samples due to metallization, the presence of carbon and oxygen was also identified, which are common components of lignocellulosic materials such as cellulose and hemicellulose [46].

The micrographs also identified some cellulose fibrils and some crystals that correspond to the crystalline cellulose, or portions of crystallized hemicellulose or potassium residues resulting from the extraction process. Cellulose fibrils may be associated with the process of extracting hemicellulose from curaua fiber. They may be responsible for increasing the mechanical strength of the film [47].

Table 5 shows the tensile tests for hemicellulose (from the best method 1: 10% _25 °C_3 h) compared to some results for polyhydroxybutyrate (PHB) and poly(lactic acid) (PLA) films found in the literature.

Table 5 shows that the values found for the hemicellulose film presented an ultimate tensile strength of 2.22 ± 0.13 MPa, an elongation at break of 14.9 ± 2.65% and a Young′s modulus of 4.17 ± 0.36 MPa. The results show that the hemicellulose film presents low ultimate tensile strength limits and low Young′s modulus, which implies that the film has a characteristic fragile material behavior. This is expected behavior for hemicellulose, corroborating the results found in GPC, where we observed that the molar mass of hemicellulose is low, which also justifies the low mechanical resistance and hemicellulose being an amorphous polymer, presenting low packaging, and consequently, low resistance.

Comparing the results of hemicellulose with those found in the literature for polyhydroxybutyrate (PHB) and polylactic acid (PLA), we observe that hemicellulose presents a higher elongation at break (14.9 ± 2.65%) than both PHB (4.12 ± 0.41%) and PLA (3.6 ± 0.2%). This characteristic is related to the deformability of the material, indicating that hemicellulose films present a greater deformability than PHB and PLA films [48,49]. This feature is important for use in packaging applications, for example.

It is also observed that the values of ultimate tensile strength and Young′s modulus for hemicellulose are lower than those of PHB and PLA. Young′s modulus is related to the stiffness of the material; the lower the value of the modulus of elasticity, the less rigid the film will be and vice versa [49]. These data show that the hemicellulose film is more flexible than PHB and PLA films.

Figure 6 shows the results of dynamic mechanical analysis. Up to approximately 70 °C, there was a decrease in the storage modulus of hemicellulose, which was also observed in analyses of citrate–chitosan aerogel foams of cross-linked hemicellulose [52]. The glass transition temperature (T_g_) of hemicellulose occurs between 150 and 220 °C [32]. The tensile tests of the film were performed with the material in its vitreous state (below T_g_), which provided an elastic response to the mechanical stress.

Meanwhile, the loss modulus presented low values, which is expected for polymer tests at low temperatures when the material is in its vitreous (rigid) state [53].

In the temperature range of 83–90 °C, the storage modulus decreased while the loss modulus reached its maximum value. This means that, for this temperature range, the frequency of the experiment (1 Hz) was comparable to the frequency of the internal movements of the material, corresponding to its viscoelastic behavior [54].

## 4. Conclusions

In this work, we explored which parameter (time, temperature, or base concentration) had the greatest impact on the yield of hemicellulose extraction from curaua fibers. The results obtained by the factorial study indicated that only the concentration was significant; when the higher concentration of KOH was used (20% *w*/*v*), a higher yield was achieved, which may be attributed to higher KOH residues and cellulose particles in the hemicellulose samples (as identified by the thermal analysis and infrared spectroscopy). To obtain a more pure hemicellulose film (without cellulose), the lower concentration should be used (as shown by FTIR and SEM).

SEM/EDS analyses indicate no significant differences between the hemicellulose samples. We can thus affirm the methodology defined as the most adequate (concentration 10% *w*/*v*, room temperature, and time 3 h) is promising from the viewpoint of conserving resources as this methodology uses the lowest concentration of reagent, energy (in terms of heating), and time for the process. Even though the analysis performed on the hemicellulose did not identify the presence of lignin, the analysis performed on the film and the residual fibers from its production process also demonstrated the absence of lignin, which indicates that the component was removed from the fibers but was separated from hemicellulose during the centrifugation process.

As for its thermal and mechanical properties, hemicellulose was demonstrated to be a thermoplastic polymer with high stability and good mechanical properties. While it proved to be relatively malleable, the film exhibited a limited elongation at break, characterizing both a flexible and fragile material. The hemicellulose film also showed a higher percentage of elongation, lower ultimate tensile strength and lower Young′s modulus compared to other natural polymers. This means that hemicellulose showed greater deformation and less rigidity, being more flexible, which is a good result for certain applications of the polymer, for example, for its use as a film.

## Figures and Tables

**Figure 1 polymers-14-02999-f001:**
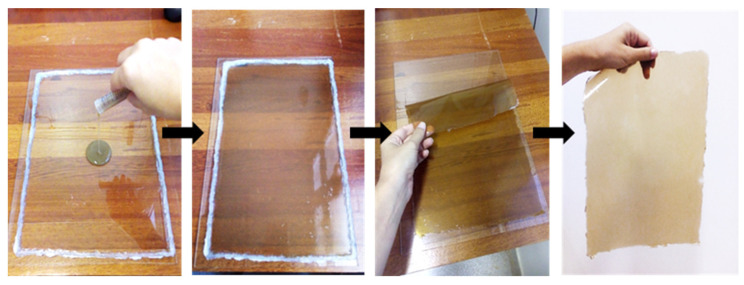
Hemicellulose films obtained by water casting.

**Figure 2 polymers-14-02999-f002:**
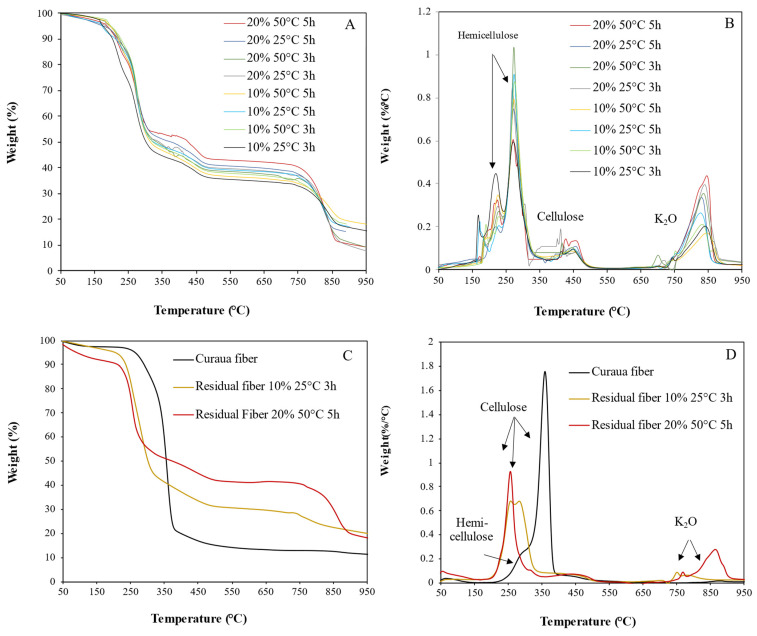
(**A**) TG curves and (**B**) DTG curves of the hemicellulose samples obtained by each extraction method; (**C**) TG curves and (**D**) DTG curves for extracted fibers subjected to the two most extreme methods of experimental design.

**Figure 3 polymers-14-02999-f003:**
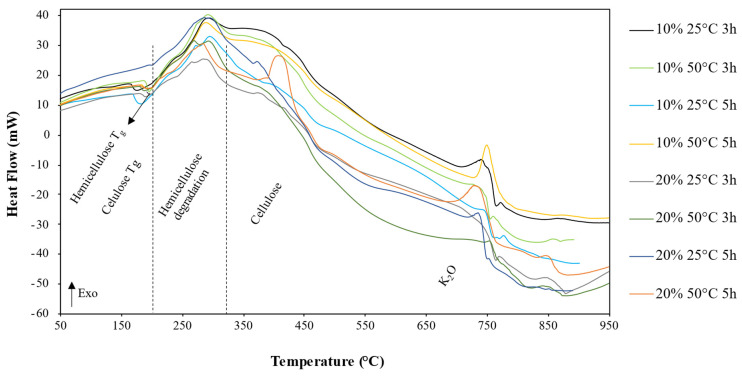
DSC curves of hemicellulose samples obtained by each method.

**Figure 4 polymers-14-02999-f004:**
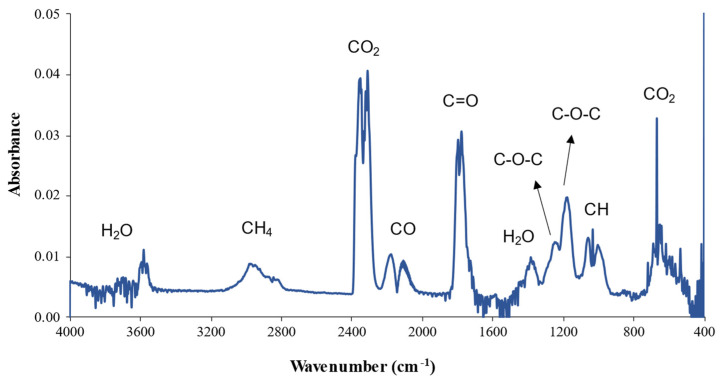
FTIR spectrum of the degradation products obtained at 311°C for hemicellulose prepared by method 1 (10%_25 °C_3 h).

**Figure 5 polymers-14-02999-f005:**
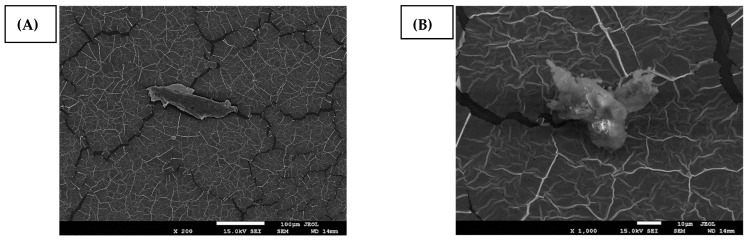
(**A**) Scanning electron microscopy of hemicellulose film at 200× magnification, indicating inorganic particles; (**B**) with 1000× magnification, indicating probable potassium residues resulting from the extraction process.

**Figure 6 polymers-14-02999-f006:**
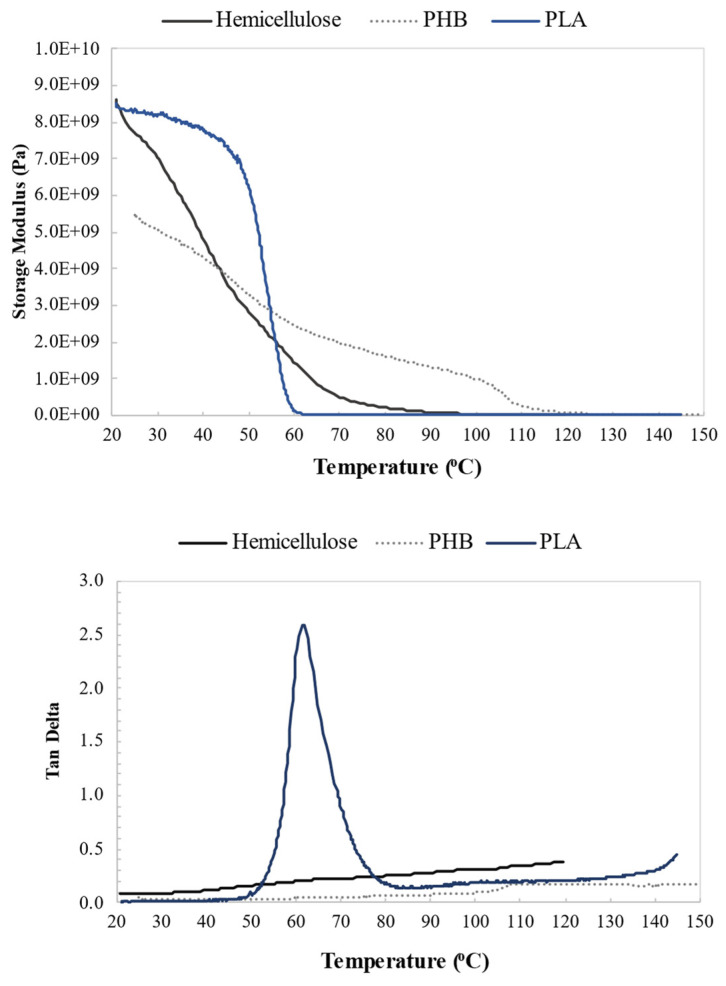
DMA curves of the hemicellulose film.

**Table 1 polymers-14-02999-t001:** The 2^3^ factorial design resulting in 16 experiments.

Level	KOH Solution Concentration (% *w*/*v*)	Temperature (°C)	Time (h)
−	10	25	3
+	20	50	5

**Table 2 polymers-14-02999-t002:** Design matrix with hemicellulose yield and values of main effects with their interactions.

Method	Concentration (% *w*/*v*)	Temperature (°C)	Time (h)	Yield (%)	Standard Deviation	Factors	Values Effects and Interactions	t 95% *
1	10	≈25 °C	3	16.03	±0.03	A	1.50	>0.24
2	20	≈25 °C	3	18.11	±0.30	C	0.30	>0.24
3	10	50	3	14.22	±0.14	T	0.02	<0.24
4	20	50	3	15.31	±0.01	t	0.11	<0.24
5	10	≈25 °C	5	13.17	±0.04	CT	−0.03	<0.24
6	20	≈25 °C	5	18.39	±0.22	Ct	0.16	<0.24
7	10	50	5	15.97	±0.15	Tt	0.23	<0.24
8	20	50	5	20.86	±0.02	CTt	0.02	<0.24

A = average; C = concentration; T = temperature; t = time; CT = interaction between concentration and temperature factors; Ct = interaction between concentration and time factors; Tt = interaction between temperature and time factors; CTt = interaction between concentration, temperature, and time factors; * The value to the Student’s t distribution with eight degrees of freedom for 95% confidence (2.31) multiplied by error (0.11) establishes 0.24 as the minimum value for the effects to be considered statistically significant.

**Table 3 polymers-14-02999-t003:** Thermal degradation for hemicellulose extracted by the different methods.

Method/Test Conditions		Thermal Stability (°C)	Hemicellulose			Cellulose		
			Peak 1 (°C)	Peak 2 (°C)	Weight Loss at Peaks (%)	Peaks (°C)	Weight Loss (%)	Residue at 900 °C (%)
1	10%_25 °C_3 h	216	220	272	48	448	7	17
2	20%_25 °C_3 h	222	226	269	43	459	11	10
3	10%_50 °C_3 h	231	230	276	43	446	8	18
4	20%_50 °C_3 h	233	229	274	46	455	10	11
5	10%_25 °C_5 h	235	229	272	43	453	7	17
6	20%_25 °C_5 h	234	220	273	41	457	9	15
7	10%_50 °C_5 h	224	227	274	49	444	8	19
8	20%_50 °C_5 h	220	225	272	42	455	11	10

**Table 4 polymers-14-02999-t004:** The absorbance of identified components in the higher intensity spectrum obtained for each sample of hemicellulose in its degradation range.

Method		H_2_O 3620–3500 cm^−1^	CH_4_ 3150–2740 cm^−1^	CO_2_ 2400–2240 cm^−1^	CO 2230–2000 cm^−1^	C=O 1850–1680 cm^−1^	H_2_O 1500–1340 cm^−1^	C-O-C 1250 cm^−1^	CO_2_ 650 cm^−1^
1	10%_25 °C_3 h	0.01	0.01	0.04	0.01	0.03	0.01	0.01	0.03
2	20%_25 °C_3 h	0.03	<0.01	0.07	<0.01	0.03	0.01	0.00	0.03
3	10%_50 °C_3 h	0.10	0.01	0.06	0.02	0.11	0.08	0.26	0.08
4	20%_50 °C_3 h	0.03	<0.01	0.15	0.01	0.03	0.01	<0.01	0.08
5	10%_25 °C_5 h	0.09	0.01	0.05	0.02	0.08	0.07	0.01	0.06
6	20%_25 °C_5 h	<0.01	<0.01	0.02	<0.01	<0.01	<0.01	<0.01	0.01
7	10%_50 °C_5 h	0.02	0.01	0.07	0.01	0.20	0.10	<0.01	0.04
8	20%_50 °C_5 h	<0.01	<0.01	<0.01	<0.01	<0.01	<0.01	<0.01	<0.01

**Table 5 polymers-14-02999-t005:** Mechanical properties of hemicellulose film compared to PHB and PLA films.

Properties	Hemicellulose Film	Polyhydroxybutyrate Film	Polylactic Acid Film
Ultimate Tensile Strength (MPa)	2.22 ± 0.13	20.87 ± 0.85 ^A^ 48.8 ± 5.3 ^B^	58.0 ± 2.8 ^C^ 38.5 ± 2.3 ^D^
Young′s Modulus (MPa)	4.17 ± 0.36	801.30 ± 24.46 ^A^ 1670 ± 50 ^B^	2240 ± 0.04 ^C^ 1150 ± 100 ^D^
Elongation Percentage, ɛ (%)	14.9 ± 2.65	4.12 ± 0.41 ^A^ 2.0 ± 0.4 ^B^	3.6 ± 0.2 ^C^ 1.5 ± 0.3 ^D^

^A^ (Marina P. Arrieta, López, Hernández, and Rayón, 2014) [50]; ^B^ (Giaquinto et al., 2017) [49]; ^C^ (Marina P. Arrieta et al., 2014) [2]; ^D^ (Jamshidian et al., 2012) [51].

## Data Availability

Not applicable.

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
