# Peer review of "Hemicellulose Films from Curaua Fibers (Ananas erectifolius): Extraction and Thermal and Mechanical Characterization"

_polymers, 2022, doi:10.3390/polym14152999_

Round 1

Reviewer 1 Report

The manuscript is very well written and structured. The study is thoroughly conducted. The results are very valuable to the scientific community and the reviewer is pleased to recommend it for publication.

Author Response

Dear reviewer,

Thank you very much for your comment. A final English language and style was checked by our co-autor Dr. Anastasia Elias, as a native speaker.

Thank you,

Reviewer 2 Report

Dear Authors,
I found your manuscript valuable and of high potential interest to readers. However, below, please, see several remarks to your manuscript:
- line 80-81, where you provided the aim of the research; in my opinion, you should tune the title of the manuscript (by adding for example information that you are trying to characterize the influence of production parameters of hemicellulose films) to make it closer to the aim and conclusions.
- I think the fig. 2 is unuseful and can be removed since the detailed description of sample dimensions is provided in lines 179-180.
- line 190, where the information about film production is given - please add the information about the temperature of molding.
- line 201: in the sentence "...is given in grams for each experiment" the "in grams" can be removed, since you said about this in line 199.
- line 205 (table 2) in my opinion the column titled "Average yield (g)" can be removed, since more useful is data titled "Yield (%)".
- line 235-236: one of "its crystalline structure" should be removed.
- line 263 - please add indicators A, B. C and D to the plots, because now is hard to find the proper plot and connect it to the right description.
Best regards!

Author Response

Dear Reviewer,

Below, we respond point by point to your comments. Besides that, we performed a complete the English checking.

- line 80-81, where you provided the aim of the research; in my opinion, you should tune the title of the manuscript (by adding for example information that you are trying to characterize the influence of production parameters of hemicellulose films) to make it closer to the aim and conclusions.

Answer: We changed the title to “Hemicellulose films from curaua fibers (Ananas erectifolius): extraction and thermal and mechanical characterization“ , to adjust the objective and conclusions of the paper.

- I think the fig. 2 is unuseful and can be removed since the detailed description of sample dimensions is provided in lines 179-180.

Answer: We removed Figure 2 from the manuscript according to your recommendation. As a result, all Figure numbers were changed.

- line 190, where the information about film production is given - please add the information about the molding temperature.

Answer: We molded at 175oC. We included this information in the manuscript.

- line 201: in the sentence "...is given in grams for each experiment" the "in grams" can be removed since you said about this in line 199.

Answer: As recommended, we removed “in grams” from the manuscript.

- line 205 (table 2) in my opinion the column titled "Average yield (g)" can be removed, since more useful is data titled "Yield (%)".

Answer: We worked on Table 2 and removed the “Average yield (g)"

- line 235-236: one of "its crystalline structure" should be removed.

Answer: Thank you! We did it.

- line 263 - please add indicators A, B. C and D to the plots, because now is hard to find the proper plot and connect it to the right description.

Answer: Thank you! We add the letters to the plots.

Reviewer 3 Report

Review of the manuscript entitled ‘Hemicellulose films from curaua fibers (Ananas erectifolius)’

 The study aims at the development of an optimized method to extract hemicellulose from curaua fibers. The influence of three main factors of the hemicellulose extraction procedure – namely KOH concentration, temperature and time – on both the yield and characteristics of the obtained material are investigated. Furthermore, after the film formation, the main properties of the material, including mechanical properties and morphology are characterized. The study explores the greatest impact of different paramters on the yield of hemicellulose extraction from curaua fibers. KOH is determinant is the fabrication of the biopolymer. Moreover, the comparison with others biopolymers is also significant.

The manuscript is well written and deserves to be published. The characterization is complete with methodology.

Author Response

Dear reviewer,

Thank you very much for your comments regarding the importance of our study. We think this study should contribute for the further hemicellulose development as a natural polymer.

Best Regards,